# FEM-GAN: A Physics-Supervised Deep Learning Generative Model for Elastic Porous Materials

**DOI:** 10.3390/ma16134740

**Published:** 2023-06-30

**Authors:** Albert Argilaga

**Affiliations:** MOE Key Laboratory of Soft Soils and Geoenvironmental Engineering, Zhejiang University, Hangzhou 310058, China; argilaga@zju.edu.cn

**Keywords:** GANs, FEM-GANs, generative modeling, physics-supervised, elasticity, X-ray CT, sacrifice template

## Abstract

X-ray μCT imaging is a common technique that is used to gain access to the full-field characterization of materials. Nevertheless, the process can be expensive and time-consuming, thus limiting image availability. A number of existing generative models can assist in mitigating this limitation, but they often lack a sound physical basis. This work presents a physics-supervised generative adversarial networks (GANs) model and applies it to the generation of X-ray μCT images. FEM simulations provide physical information in the form of elastic coefficients. Negative X-ray μCT images of a Hostun sand were used as the target material. During training, image batches were evaluated with nonparametric statistics to provide posterior metrics. A variety of loss functions and FEM evaluation frequencies were tested in a parametric study. The results show, that in several test scenarios, FEM-GANs-generated images proved to be better than the reference images for most of the elasticity coefficients. Although the model failed at perfectly reproducing the three out-of-axis coefficients in most cases, the model showed a net improvement with respect to the GANs reference. The generated images can be used in data augmentation, the calibration of image analysis tools, filling incomplete X-ray μCT images, and generating microscale variability in multiscale applications.

## 1. Introduction

The use of nanoporous sacrifice templates is a common technique to obtain negative pore metamaterials (e.g., [1]). Once the filling material is in place, the template can be removed, hence the “negative porosity” or “sacrifice template” designations. The fabrication of porous materials by a sacrificial template using various biological, chemical or synthetic templates has numerous applications such as thermal insulation [2], water desalinization [3], and electronics [4]. Many applications are related to bone scaffold substitution, which focus on the mechanical properties. Kim et al. use photocrosslinked poly(propylene fumarate) (PPF)- and diethyl fumarate (DEF)-constructed scaffolds to study the mechanical stiffness of substrate materials, pore size, and channel geometry [5]. Wu et al. studied porous mesopore–bioglass (MBG) scaffolds with a focus on the material’s inherent brittleness, high degradation, and surface instability [6]. Hydraulic properties have been the main focus in other studies; Reinwald et al. [7] investigated the interconnectivity and permeability of scCO${}_{2}$-foamed polymer scaffolds fabricated with supercritical fluid technology. The polymer was subjected to a supercritical fluid (exceeding its critical temperature and pressure), thereby creating the sacrifice template with this technique. Supercritical fluid technology has proved effective in the fabrication of materials with tailored structural properties, such as porosity and pore size. In Reinwald et al. [7] the scaffolds were fabricated with Poly(lactic-co-glycolic acid) (PLGA) with average molecular weights of 37 kDa, 53 kDa, and 109 kDa with the aim of modulating pore connectivity and pore window sizes. SEM images (Figure 1) show results with the typical characteristics of such material; convex pores connected through relatively small pore windows. The authors concluded that larger PLGA molecular weights result in larger pore sizes and windows, thus increasing permeability. Although the study focused on pore and window sizes only, it is apparent that the different molecular weights may probably affect the mechanical properties as well.

Template materials are often of a biological nature or have biological-like geometry, e.g., macroporous crystals with urchin skeletal sponge-like morphologies produced by templating with a sponge-like polymer membrane [8], as well as porous biomaterial with structural and mechanical properties similar to cancellous bone prepared using biogenic hydroxyapatite and glass [9]. The use of geomaterials such as sand for the sacrificial template remains unstudied. Such a ubiquitous material could bring economic and environmental advantages compared to its chemical or biological counterparts. In addition, the use of granular soils as a template could allow for the in situ fabrication of materials in scenarios such as soil improvement in foundations [10], constructed barriers in nuclear waste repositories [11], etc.

The advent of deep learning and data science has profoundly impacted some fields, e.g., image recognition [12]. Their success can be explained by their ability to totally replace human intervention in tasks such as classification. In some cases, the application of deep learning techniques in more traditional scientific disciplines can unlock unseen capabilities (e.g., control theory in geomechanics [13]). In other cases, deep learning has been limited to fitting problems, e.g., in the prediction of pile bearing capacity [14], slope staility [15], or rupture distances and capillary forces in wet granular media [16]. Fitting problems can be enriched with physics and thermodynamics supervision as well [17]. In those applications, deep learning usually results in vastly improved accuracy metrics. Nevertheless, where deep learning is especially suited is in more complex applications such as generative models.

In geomechanics and material science, many works address the development of generative models. Those are intended to create materials with a given set of properties while fulfilling classical physical and topological restrictions. Notable examples of those generative models include Markovian theory [18], cross-correlation [19], Fourier transform [20], level-set [21], spherical-harmonic-based principal component analysis [22], and the Inwards Packing Method [23]. Such approaches lack the generalization capability of deep learning and often require prior knowledge of material characteristics or probability distributions of the target materials. Supervised deep learning predictive [24] and generative [25] models have emerged in recent years with applications in material science. Supervised learning models are inherently limited by the loss function, upon which they rely to evaluate performance. In contrast, unsupervised learning models have a higher freedom to learn features but lack a metric to evaluate the posterior error. To address those limitations, physics information has been recently used to improve machine learning training in nonlinear partial differential equations [26]. Similarly, regularization in canonical problems in the form of physical supervision showed good results [27,28]. The use of pseudophysics information (permeability in porous media inferred from fractal metrics) has shown positive results in the generation of X-ray μCT images [29].

The objective of this work was to develop a physics-supervised deep learning model for the generation of X-ray μCT images of a porous material. Generative adversarial networks [30] (GANs for short) have been retained as the base deep learning method, and they were enriched with physics supervision consisting of the elastic coefficients from FEM simulations.

The rest of the paper is organized as follows: In Section 2, the method is developed with enough detail for reproducibility, including a link to the proposed code. In Section 3, the results are presented. Validation is provided in Section 4, again including a link to the validation code to allow for full reproducibility. The paper ends with conclusions in Section 5.

## 2. Method

In this section, the base material consisting of Hostun sand is described, as well as the corresponding oedometric test. The numerical problem to be solved with FEM and numerical homogenization is also made available. Finally, the GANs deep learning approach and the physics-informed FEM-GANs resulting from the combination of GANs and FEM are presented.

### 2.1. Hostun Sand Oedometric Tets and X-ray μCT

Hostun sand was used as sacrifice template in this work. Hostun sand has been long used as a study geomaterial [31,32,33]. Hostum sand of quartzitic nature was obtained from a quarry close to Hostun, France. Due to the crushing genesis of the grains, their shape was mostly angular. This characteristic is unusual in sacrifice templating materials and could potentially allow the design of a wider range of porosities and mechanical properties because of the higher bulk shear strength. The sand specimens used had a narrow size distribution and a median grain diameter $d_{50}=338\;$μm; maximum and minimum void ratios were established at $e_{min}=0.606$ and $e_{max}=0.928$. Given the target use as a sacrifice template, the negative porosity $n_{st}$ can be defined as the porosity of the moulded material through Equation (Equation 1):(1)$$n_{st}=1-n,$$
where *n* is the porosity of the original sand. This resulted in minimum and maximum porosities: $n_{st-min}=0.519$ and $n_{st-max}=0.623$. The negative void ratio $e_{st}$ is defined as the void ratio of the moulded material though Equation (Equation 2):(2)$$e_{st}=\frac{1}{\left(e+1\right)\left(1-\frac{1}{e+1}\right)},$$
where *e* is the void ratio of the original sand. This resulted in minimum and maximum void ratios: $e_{st-min}=1.079$ and $e_{st-max}=1.653$.

An oedometric compression test on a compacted Hostun sand sample was used in the following. An initial medium-dense density was given to the sample for the test. Loading was displacement controlled using the oedometer setup in the tomograph in Laboratoire 3SR, Grenoble, which was developed by Zeynep Karatza [34]. The description of the full experimental procedure and X-ray μCT image acquisition can be found in the PhD thesis of Max Wiebicke [35]. For the present study, the image at point 4 of the oedometric loading (see [35], page 103, Figure 5.7) was used. This point was part of a charge–discharge branch during the oedometric test with axial stress $\sigma_{z}=500$ kPa and axial strain $\epsilon_{z}=0.019$. The selection of this point obeyed the following criteria: (1) the relatively low axial stress minimized the presence of grain breakage, (2) the charge–discharge branch was essentially elastic, and (3) the axial stress was high enough to assume that most of the particles were in contact. The 3D X-ray μCT image at the described loading point is shown: isosurface with threshold 50% (Figure 2a) and orthogonal slice planes (Figure 2b).

The oedometric cylinder was cropped by removing a length of 100 pixels from the top and bottom boundaries and 80 pixels from the cell perimeter to avoid any boundary effects. The remaining space was then randomly sampled with a $64^{3}$ pixel window resulting in a database with 1776 images.

### 2.2. Homogenized Elastic Coefficients from X-ray μCT Images

To obtain the mechanical properties from the X-ray μCT images, a method based on image segmentation and continuum mechanics was proposed. For the sake of reproducibility, the full code is provided in the external repository with DOI: 10.6084/m9.figshare.23280860 (posted on 2 June 2023). First, a histogram-based segmentation was applied to the X-ray μCT image to distinguish between solid grains and voids. Initially, the image was in black and white, with white level values proportional to X-ray absorption, but, to simulate the molded material instead of the sacrifice template, higher white levels were considered void and vice-versa. Image bit depth was 16, which translated into 65,535 levels of white; a white level threshold of 30,000, or 45.78% in a normalized scale, was selected. This threshold was approximately equidistant from the corresponding solid and void white level peaks (Figure 3).

The resulting topology was meshed using the open source code *iso2mesh* [36] based on the Surface Mesh Extraction Utility (CGAL 5.0) and available at http://iso2mesh.sf.net (accessed on 25 July 2022).

### 2.3. FEM

A finite element model was used to numerically solve the mechanical boundary value problem (BVP). The strong formulation of the elasticity equations is presented in Equations (Equation 3)–(Equation 5):(3)divσ=0inΩ,
(4)$$\sigma =c:\epsilon (\vec{u})\;in\;\Omega ,$$
(5)$$\sigma \cdot \vec{n}=\vec{t}\;in\;\partial \Omega_{t},$$
where Ω is the BVP domain, $\partial \Omega_{t}$ is the domain boundary where traction is applied, σ is the Cauchy stress, *c* is a fourth order stiffness tensor, $\epsilon (\vec{u})$ is the strain field function of the displacement field $\vec{u}$, $\vec{n}$ is the normal to the boundary, and $\vec{t}$ is the applied boundary traction. The problem is made complete with boundary conditions. The first step to obtaining the weak formulation is to write the virtual power formulation of the equilibrium. This is done by scalarly multiplying Equation (Equation 3) by a virtual velocity field $\vec{\omega }$ and integrating it over the domain Ω, as achieved using Equation (Equation 6):(6)$$\int_{\Omega }\mathrm{div}\;\sigma \cdot \vec{\omega }\;dA=0.$$

Secondly, integrating by parts and using Ostrogradsky’s divergence theorem obtains Equation (Equation 7):(7)$$\int_{\Omega }\sigma :\nabla \vec{\omega }\;dA=\int_{\partial \Omega }\left(\sigma \cdot \vec{\omega }\right)\cdot \vec{n}\;dS,$$
where ∂Ω is the complete boundary of the domain Ω. Following the standard procedure, i.e., considering the symmetry of σ in the left hand side integral, and then linearly manipulating the right hand side yields Equation (Equation 8):(8)$$\int_{\Omega }\sigma :\epsilon \left(\vec{\omega }\right)\;dA=\int_{\partial \Omega }\left(\sigma \cdot \vec{n}\right)\cdot \vec{\omega }\;dS.$$

By setting the test field acting on the part of the boundary condition without applied traction to zero and substituting the expression in the right hand side integral by Equation (Equation 5), we obtain the weak formulation Equation (Equation 9):(9)$$\int_{\Omega }\sigma :\epsilon \left(\vec{\omega }\right)\;dA=\int_{\partial \Omega_{t}}\vec{t}\cdot \vec{\omega }\;dS.$$

The weak formulation is finally discretized according to Galerkin’s approach, thus resulting in Equation (Equation 10):(10)Ku=f,
where *u* is the nodal displacement vector, *f* is the nodal force vector, and *K* is the global rigidity matrix. The problem is then about minimizing the residual force *R* using Newton–Raphson techniques in the present case, or other quasi-Newton techniques if *K* was to be obtained by numerical differentiation [37], as demonstrated in Equation (Equation 11):(11)$$R=\int_{\Omega }B^{T}\sigma \;dA-f,$$
where *B* is the deformation matrix. Since it is a FEM problem defined in a BVP domain Ω (the solid fraction of the material created with sacrifice templating), the next section details the homogenization of the problem in an elementary cell (the entire volume, including the solid fraction and voids).

### 2.4. Homogenization

Define the average stress tensor $\langle \sigma \rangle$ in an elementary cell *Y* as the following:(12)$$\langle \sigma \rangle =\frac{1}{\left|Y\right|}\int_{\Omega }\sigma \;dA.$$

The purpose is now to determine the balance equation of the homogenized medium (see e.g., [38,39]). The previous virtual power formulation of the macroscopic equilibrium reads as follows:(13)$$\forall \vec{\omega },\vec{\omega }=0\mathrm{on}\partial \Omega \;,\;-\int_{\Omega }\langle \sigma \rangle :\epsilon \left(\vec{\omega }\right)\;dA=0,$$
which proves that the divergence of the average stress is zero in the domain. Taking into account the constitutive Equations (Equation 3) to (Equation 5) results in the weak formulation of the self-balanced problem in the elementary cell, i.e., given $\epsilon \left(\vec{u}\right)$ in ∂Y, find $\vec{u}\left(\vec{x},\vec{y},\vec{z}\right)$ such that:(14)$$\forall \vec{u}\;,\;\int_{\Omega }\left(c^{H}:\epsilon \left(\vec{u}\right)\right):\epsilon \left(\vec{\omega }\right)\;dA=0,$$
where $c^{H}$ is the homogenized fourth order stiffness tensor in the elementary cell *Y*. Therefore, the macroscopic constitutive equation reads as follows: (15)$$\langle \sigma \rangle =c^{H}:\epsilon \left(\vec{u}\right),$$
or in matrix notation, it reads as the following:(16)$$\left(\begin{array}{c} {\langle \sigma \rangle }_{11}\\ {\langle \sigma \rangle }_{22}\\ {\langle \sigma \rangle }_{33}\\ {\langle \sigma \rangle }_{12}\\ {\langle \sigma \rangle }_{23}\\ {\langle \sigma \rangle }_{13} \end{array}\right)=\left(\begin{array}{cccccc} c_{1111}^{H} & c_{1122}^{H} & c_{1133}^{H} & c_{1112}^{H} & c_{1123}^{H} & c_{1113}^{H}\\ c_{2211}^{H} & c_{2222}^{H} & c_{2233}^{H} & c_{2212}^{H} & c_{2223}^{H} & c_{2213}^{H}\\ c_{3311}^{H} & c_{3322}^{H} & c_{3333}^{H} & c_{3312}^{H} & c_{3323}^{H} & c_{3313}^{H}\\ c_{1211}^{H} & c_{1222}^{H} & c_{1233}^{H} & c_{1212}^{H} & c_{1223}^{H} & c_{1213}^{H}\\ c_{2311}^{H} & c_{2322}^{H} & c_{2333}^{H} & c_{2312}^{H} & c_{2323}^{H} & c_{2313}^{H}\\ c_{1311}^{H} & c_{1322}^{H} & c_{1333}^{H} & c_{1312}^{H} & c_{1323}^{H} & c_{1313}^{H} \end{array}\right)\left(\begin{array}{c} \epsilon_{11}\left(\vec{u}\right)\\ \epsilon_{22}\left(\vec{u}\right)\\ \epsilon_{33}\left(\vec{u}\right)\\ \epsilon_{12}\left(\vec{u}\right)\\ \epsilon_{23}\left(\vec{u}\right)\\ \epsilon_{13}\left(\vec{u}\right) \end{array}\right).$$

By considering a homogeneous strain field $\epsilon \left(\vec{v}\right)$ applied on the boundary of the elementary cell ∂Y and the zero divergence of the average stress within the domain, the homogenized coefficients of the stiffness tensor can be computed, as shown in Equation (Equation 17):(17)$$c_{ijkl}^{H}=\frac{{\langle \sigma \rangle }_{kl}}{\epsilon_{ij}\left(\vec{v}\right)}.$$

On the standard energy-based assumptions of symmetries for the elastic tensor *c*, namely, $c_{ijkl}=c_{klij}$, it is classical (see [40] or [41]) to prove that $c^{H}$ satisfies them; therefore, only those are regarded in the following Equation (Equation 18):(18)$$c_{ijkl}^{H}=c_{klij}^{H}.$$

Only 9 out of the initial 36 $c_{ijkl}^{H}$ coefficients (Equation (Equation 16)) are needed to define general anisotropic materials, and the problem is reduced to the solving of the “kite” coefficients in Equation (Equation 19):(19)$$\left(\begin{array}{c} {\langle \sigma \rangle }_{11}\\ {\langle \sigma \rangle }_{22}\\ {\langle \sigma \rangle }_{33}\\ {\langle \sigma \rangle }_{12}\\ {\langle \sigma \rangle }_{23}\\ {\langle \sigma \rangle }_{13} \end{array}\right)=\left(\begin{array}{cccccc} c_{1111}^{H} & c_{1122}^{H} & c_{1133}^{H} & 0 & 0 & 0\\ c_{2211}^{H} & c_{2222}^{H} & c_{2233}^{H} & 0 & 0 & 0\\ c_{3311}^{H} & c_{3322}^{H} & c_{3333}^{H} & 0 & 0 & 0\\ 0 & 0 & 0 & c_{1212}^{H} & 0 & 0\\ 0 & 0 & 0 & 0 & c_{2323}^{H} & 0\\ 0 & 0 & 0 & 0 & 0 & c_{1313}^{H} \end{array}\right)\left(\begin{array}{c} \epsilon_{11}\left(\vec{v}\right)\\ \epsilon_{22}\left(\vec{v}\right)\\ \epsilon_{33}\left(\vec{v}\right)\\ \epsilon_{12}\left(\vec{v}\right)\\ \epsilon_{23}\left(\vec{v}\right)\\ \epsilon_{13}\left(\vec{v}\right) \end{array}\right).$$

Further simplifications follow for cross-anisotropic, or ultimately isotropic materials (only two coefficients). The hypothesis of general anisotropy is retained in the following for the homogenized medium. Note that *c* in Euqation (Equation 4) represents the skeleton material and is considered to be isotropic.

### 2.5. Generative Adversarial Networks (GANs)

GANs, which were proposed by Goodfellow [42], involve a deep learning approach in which the discriminator is a neural network. Numerous GAN variants have been proposed since its initial introduction in 2017, including Least Squares Generative Adversarial Networks (LSGANs) by Mao et al. [43] and Wasserstein Generative Adversarial Networks (WGANs) by Arjovsky et al. [44], among others, and they play an essential role in the generation of artificial images, audio signals, and others. GANs can be used as generative models [45] and as predictive models as well [46,47].

In GANs, a generator (*G*) consisting of a deep neural network generates samples to mimic the ones of the training distribution, while the discriminator (*D*) gives the probability of the samples coming from the training dataset rather than from the generator. The objective of the training for *G* is to maximize the probability of *D* making a mistake. The unsupervised adversarial game between the generative and discriminative networks continues until Nash equilibrium is reached. Then, *G* can be fed with random noise from the distribution $p_{z}\left(z\right)$ to generate synthetic samples. This two-sided minimax game can be written using the value function $v\left(D,G\right)$ as follows:(20)$$\min_{G}\max_{D}v\left(D,G\right)=E_{x_{data}\left(x\right)}\left[\log\left(D\left(x\right)\right)\right]+E_{Z_{z}\left(z\right)}\left[\log\left(1-D\left(G\left(z\right)\right)\right)\right],$$
where D(x) is the probability of *x* coming from the data rather than the *G* distribution $p_{G}$. *D* and *G* are simultaneously trained in order to assign the correct tag to data samples and *G* samples (the discriminator) and to minimize $\log\left(1-D\left(G\left(z\right)\right)\right)$ (the generator). The generator loss is then defined as follows:(21)$$v_{G}\left(D,G\right)=-E_{Z_{z}\left(z\right)}\left[\log\left(D\left(G\left(z\right)\right)\right)\right].$$

The GANs diagram is presented (Figure 4). The diagram shows two paths of samples going through the discriminator; once the prediction is made, the image nature (fake or real) is revealed to compute the loss and to update *D* and *G* with errors and gradients.

While in GANs, the networks are usually constituted by multilayer perceptrons (fully connected layers), in deep convolutional generative adversarial networks (DCGANs), convolution filters learn the features in each layer. This series of convolution filters are used similarly to neural networks in the visual system of humans and animals. A DCGAN architecture with features adopted from [29,48] was used in the following, but the method is still referred to as a GAN for short. In particular, the following network design features that improve training stability in larger image sizes were used:An all-convolutional approach replaced deterministic spatial pooling filters. Convolutional filters are very effective at extracting image features in the different layers without using excessive memory.Fully connected layers on top of convolutional features were eliminated. The only fully connected layers were found in the generator input and discriminator output.Batch normalization was applied after the convolution layers to stabilize learning. However, due to its tendency to create instability when applied to all layers, it was not applied in the generator output or the discriminator input [49].

The architecture of the generator network is given (Table 1). Rectifier linear unit (ReLU) activation functions were used after the convolutions. The generator’s output used a hyperbolic tangent (tanh) activation to obtain a continuous value in the interval [−1,1] that matched the standardized white level of the images.

The architecture of the discriminator network was almost a mirrored image of the generator one (Table 2). Leaky rectifier linear unit activation functions (Leaky ReLU) were used in the discriminator after the convolutions. Leaky ReLU is a modification of ReLU activation functions with a slight slope given to the function before the activation threshold; this avoids zero gradient issues during training. In contrast to the generator, the scalar output of the discriminator needs to be a probability in the interval [0,1]; a sigmoid activation function was used. Dropout was used after the discriminator’s activations as a regularization to improve generalization and training stability.

### 2.6. Finite Element Method-GANs (FEM-GANs)

Most of the literature on physics-informed machine learning methods is devoted to the enrichment of supervised schemes using physics laws (e.g., Masi et al. [50]). In particular, vanilla neural networks (multilayer perceptron with one single hidden layer) gained great popularity among the scientific community because of their applicability to any predictive problem, and they usually outperformed the accuracy of previous predictions. Physical restrictions can be added to the loss function as penalty terms, thus requiring minor or no additional modification of the neural network.

In unsupervised learning, from an outside observer’s eye, there is no supervising metric, because the output of the network is not a variable defined in the database. Nevertheless, when looking at the individual components of a GANs scheme (generator and discriminator networks), the generator follows a supervised learning in which the supervision is performed by the discriminator network. In other words, the discriminator provides the posterior quality metric used for supervision. This globally unsupervised–locally supervised architecture allows for adding additional supervision modules in parallel to the discriminator. In this work, a physics-based evaluation of elastic properties was used (Figure 5).

The main differences between the GANs neural network discriminator and the physics-based discriminator are in the initial knowledge and learning capacity. The neural network discriminator starts the learning process with zero knowledge, identical to the generator, and both synchronously update their weights and biases during learning. On the other hand, the physics-based discriminator does not have learning capability; in this sense, it is similar to the error computation in a supervised learning model.

The physics-based discriminator is composed of the following stages: (1) First, there is histogram-based segmentation of the images and 3D meshing of the topological surface using tetrahedral elements. (2) Second, there is the building of the BVP by imposing boundary displacements, thus providing material elastic properties, solving the FEM problem in all DoF (6 in 3D), and homogenizing the results to obtain the full matrix of elastic coefficients. (3) Third, the nonparametric Wilcoxon rank sum test is used, and the *p*-values of the nine “kite” coefficients are obtained. Those give a probability of the generated values belonging to the distribution of the database. During training, the neural network discriminator is fed with batches of images, one batch per iteration, through all the images in the database, thus defining one epoch. For each batch of real images, one batch of fake images (coming from the generator) is fed to the discriminator neural network. In the case of the FEM-informed discriminator, the 1776 images from the training database only needed to be segmented, meshed and solved with the FEM once. Since this has a significant computational cost compared to the total, the values of the elastic coefficients $c_{ijkl}^{H}$ were stored as a database of elastic properties (Figure 6).

This computation was performed before any neural network training took place. Subsequently, at each iteration, the entire elasticity database was retrieved and used to perform nonparametric Wilcoxon rank sum tests on the elasticity values coming from generated images (Figure 7). This avoided the redundant computation of images previously used in past epochs.

Compared to the neural network discriminator, the FEM-informed discriminator has information on the entire database at all times, while the neural network counterpart is restricted to the batch size. This key difference between the two discriminators explains why the FEM-informed one can only be activated after a pre-training of the neural network one. If the FEM-informed discriminator was activated at the beginning when the generator could not yet produce good images, it would easily reject all the images without providing any useful learning gradient. This phenomenon, known as learning imbalance, is also common in noninformed GANs.

The *p*-values obtained from the nonparametric tests were then used to compute a loss, in this case, to update the weights and biases of the generator neural network. Two loss functions were considered in this work. In the first one, the new informed loss $v_{G}$ is computed as the average of the classical loss and the FEM-informed loss:(22)$$v_{G}\left(D,D_{FEM},G\right)=-0.5E_{Z_{z}\left(z\right)}\left[\log\left(D\left(G\left(z\right)\right)\right)+\log\left(D_{FEM}\left(G\left(z\right)\right)\right)\right],$$
where $D_{FEM}\left(G\left(z\right)\right)$ is the *p*-value given by the FEM-informed discriminator. In the second one, the new information is not added as an independent component of the total loss, but instead, the loss is computed as the norm of the logarithms. This formulation has been shown to perform well in 2D physics-informed GANs [29].
(23)$$v_{G}\left(D,D_{FEM},G\right)=-E_{Z_{z}\left(z\right)}\sqrt{\log{\left(D\left(G\left(z\right)\right)\right)}^{2}+\log{\left(D_{FEM}\left(G\left(z\right)\right)\right)}^{2}}.$$

The code was implemented in Matlab using CUDA libraries, and computations were performed in single precision arithmetic on an NVIDIA GeForce RTX 3080 Ti GPU with 16 GB of dedicated memory and a CUDA capability of 8.6. Once the latent dimension was fixed to 128, the batch size needed to be limited to 8 because of GPU memory usage (Table 3). With the 1776 image database, this resulted in 222 iterations per epoch. The batch size of 8, which was sensibly smaller than the central limit theorem reference value (approximately 32), is the reason behind the adoption of nonparametric statistics in the FEM-informed discriminator. The rest of hyperparameters are presented in Table 3.

In this work, the FEM discriminator did not necessarily evaluate image batches at every iteration; because of its computational cost, FEM evaluation frequency was reduced to selected values. The average computation time per epoch was between 308 and 1190 s depending on FEM evaluation frequency, with a total between 4 and 16 h for a 50-epoch training of an already pretrained network. The complete code is available in the external repository with DOI: 10.6084/m9.figshare.23280860 (posted on 2 June 2023). Note that a CUDA capable GPU with a minimum of 9 GB of dedicated memory will be needed in order for the networks to fit the memory. If not enough GPU memory is available, the batch size and latent dimension can be decreased until both networks fit in the memory, but in that case, the behavior of the model may differ from the one presented here.

## 3. Results

To study the properties of the generated images, a central slice orthogonal to the direction *z* (parallel to the axis of the cylindrical sample) of the $64^{3}$ voxel images was saved at the end of each epoch for 200 epochs. The evolution of the generations showed high variability and the presence of artifacts in the first half (Figure A1) with better quality images and well-defined pore–solid boundaries in the second half (Figure A2). As stated before, to avoid a learning imbalance, the FEM-informed discriminator cannot be activated until the generator produces reasonably accurate images. Therefore, a GANs pretraining of 150 epochs was performed before any supervision was added to the model.

The 150-epoch pretrained networks (PT) were saved as a common starting point for successive trainings. A reference simulation (REF) continued from the pretrained state for an additional 50 epochs; the rest of FEM-GAN simulations were compared against the REF simulation (Figure 8).

Simulations A4 and A5 used the average loss function Equation (Equation 22) in the FEM-informed discriminator, and the images were evaluated sparingly; FEM evaluations were performed every 28 iterations in A4 and every 14 iterations in A5. The probability value obtained by the FEM-informed discriminator $D_{FEM}\left(G\left(z\right)\right)$ was carried over in the iterations without evaluation. This persistent probability strategy was based on the assumption that the probability would change along iterations in an incremental and smooth manner. In simulations B1 and B4, the same average loss was used from Equation (Equation 22), and images were evaluated once per epoch (B1) and every 28 iterations (B4). However, unlike in A4–A5, the value $D_{FEM}\left(G\left(z\right)\right)$ was ephemeral, i.e., its value was reset to one in all iterations without evaluation. A probability value of one means that the discriminator believes that the images are real and gives null loss (no updating of networks). All previous simulations were run for 50 epochs, except for A5, which halted at epoch 41 due to a meshing error; this meshing error appeared to be of physical nature because of the images containing too much noise. A restart of the learning process after epoch 41 proved to be impossible because of the noisy images. In simulations C1 and C2, the average loss was used (Equation (Equation 22)), and FEM evaluations were performed in every single iteration: C1 ran for one epoch, and C2 ran for two epochs. In simulations D1 and D2, the norm loss was used (Equation (Equation 23)), and FEM evaluations were performed in every single iteration: C1 ran for one epoch, and C2 ran for two epochs. The simulation overview can be found in Table 4.

The first step of the FEM-informed discriminator is the histogram-based segmentation, meshing, and obtaining a geometry. Geometry examples are shown for the different types: a real image (Figure 9a), a generated image after pretraining (Figure 9b), a generated image in the reference state (Figure 9c), and an image generated with FEM-GANs, as whown in simulation D1 (Figure 9d). Elastic properties corresponding to quartz were used for the raw material (stiffness tensor *c* in Equation (Equation 4)), i.e., the elastic modulus E=90 GPa and the Poisson ratio μ=0.17.

The previous BVP was 3D-meshed using tetrahedral elements and quadratic shape functions. Meshes of the different types of images are shown (Figure 10). For comparison, the four selected images correspond exactly to the ones shown in the previous figure. The mesh node count was in the order of 10,000, and the FEM routine was programmed to return an error and stop the simulation for meshes with a node count outside the interval [2850, 28,800]. This interval was imposed because meshes with very high node counts are usually indicative of noisy images, while very low node counts are indicative of an improper generation of porosity. If the FEM-GANs model was applied early in GANs training (when the generator does not yet produce good images), node count errors were encountered continuously. On the other hand, after a 150 epochs pretraining, the images were almost always within the established interval (with exceptions, see simulation A5 halting in Table 4).

The previous mesh was solved using the FEM. Because of the small strain elastic nature of the problem, an arbitrary strain magnitude could be applied in the form of a homogeneous strain field to the boundary of the image domain (see Equation (Equation 17)). In the 3D, six independent strain components were required to obtain all the elasticity coefficients.

Figure 11 shows, as an example, the displacement solution in the vertical direction and reference configuration for an applied strain $\epsilon_{33}=1$. The FEM solving was the most computationally expensive step of the entire FEM-GANs code; therefore the six FEM resolutions were parallelized. To increase the performance even further, readers are encouraged to parallelize the FEM routine at a higher level (when called from the GANs code). At the time of this research, this has not been deployed because of a conflict with the mesher *iso2mesh*.

The FEM results were homogenized for each one of the stress components and strain inputs to obtain all the coefficients of the elasticity tensor $c_{ijkl}^{H}$ (36 in general). The magnitudes of the elasticity tensor coefficients were normalized to 8-bit values for graphic display (Figure 12). The figures show the typical kite shape with 9 independent (12 non-zero) coefficients in agreement with standard energy-based assumptions about symmetry. The diagonal components 11, 22, and 33 showed slight differences in magnitude that indicated a moderate material anisotropy. This anisotropy was consistent with the genesis of the sacrifice template consisting of an oedometric loading in the *z* direction.

In the previous figures, one single image of each group is shown. To validate the model, statistical analyses, including larger sample sizes, are needed. In the next section, the validation of the model was performed using both parametric and nonparametric statistics. The validation was performed on a per-component basis in which the nine “kite” coefficients were individually evaluated.

## 4. Validation

The FEM-informed discriminator in the FEM-GANs approach used Wilcoxon rank sum tests to evaluate the elastic properties of the generated images. The use of nonparametric statistics was required because of the batch size limitation. In the validation, there was no such limitation; larger quantities of images could be generated to compute parametric statistics. In the following, parametric (*t*-tests) were performed, in addition to nonparametric tests (Wilcoxon rank sum), to evaluate the fit of the generated data to the real data in terms of elastic properties.

Samples of 32 generated images with 1776 ground truth images were used for the nonparametric tests, and 100 generated images and 100 ground truth images were used for the parametric tests (Table 5). The 100 images from the database were drawn once at random for the parametric tests, and the rest were generated from the trained network at each stage (i.e., pretraining, reference, and the eight FEM-GANs cases). The testing sample was enlarged for the *t*-tests, because they require two equally-sized samples to be compared, thereby decreasing the likelihood to reach a rejection of the null hypothesis compared to the rank sum test, which uses the entire database of 1776 images as the ground truth.

Rank sum tests were performed on each individual elasticity coefficient (nine in total after dismissing symmetries) for the pretraining, reference, and the other eight simulations (Table 6). Bold values indicate a failure to reject the null hypothesis of samples with equal median. From left to right, the table presents the three diagonal components, the three out-of-axis components, and the three diagonal shear components. In the reference, the tests failed to reject the null hypothesis of equal medians in three cases (i.e., $c_{3333}^{H}$, $c_{1212}^{H}$, and $c_{2323}^{H}$), which meant that those cases could not be discarded from being real at a significance level of 5%. The number of coefficients with a failed rejection of the null hypothesis (h=0) is displayed in the last column. Simulations B4, C1, C2, D1, and D2 presented a higher count of hypothesis test results with h=0 compared to the reference, thus meaning that they are better in terms of the number of coefficients that were well reproduced. Interestingly, all hypothesis test rejections (h=0) happened for the diagonal terms (either axial or shear components), and the model failed to correctly reproduce out-of-axis coefficients (i.e., $c_{1122}^{H}$, $c_{1133}^{H}$, and $c_{2233}^{H}$).

Upon repeating the same analysis using parametric statistics (*t*-tests), simulations B4, C2, and D1 presented a higher count of hypothesis test results with h=0 (Table 7). In this case, simulation D2 matched five out of the nine coefficients by scoring good *p*-values, even in two of the out-of-axis coefficients ($c_{1133}^{H}$ and $c_{2233}^{H}$). Although B4 and D2 could reproduce out-of-axis coefficients, this came at the cost of lower *p*-values in the diagonal coefficients, thereby reducing the overall h=0 count.

Considering previous results, it is apparent that the informed FEM-GANs model could perform better than standard GANs in generating accurate elastic properties in a hypothetical material obtained with the sacrifice templating technique. Configurations B4, C2, and D1 were the only ones performing better than standard GANs in both parametric and nonparametric tests. On the other hand, simulations B4 and D2 were the only ones to score some rejections (h=0) in the out-of-axis coefficients, which could be a desirable feature depending on the material application. The author recommends the use of configuration D1 (1 additional epoch, FEM evaluations at every step, 222 FEM evaluations in total, and loss of type norm) in applications in which only diagonal coefficients are essential. In addition, the use of configuration B4 (50 additional epochs, FEM evaluations at every 8 steps, 400 FEM evaluations in total, and loss of type average) is recommended in applications in which out-of-axis coefficients are targeted.

In relying on the central limit theorem, box plots were used to compare the reference-generated images, FEM-GANs-generated (configuration D1) images, and database ground truth images in order to better explain the worst results for the out-of-axis coefficients. As in the two-sample *t*-tests, all three cases used a sample size of 100 images (Figure 13). The red line indicates the median, blue boxes enclose the interquartile range between percentiles 25 and 75, black dashes are the upper and lower adjacents, and red crosses are outliers.

While nonparametric statistics allow for testing the median and parametric statistics allow for testing the mean, box plots allow for visualizing noncentral tendency statistics such as symmetry and data spread. The first observation from the out-of-axis coefficients ($c_{1122}^{H}$, $c_{2233}^{H}$, and $c_{1133}^{H}$) is that the median was indeed improved by the FEM-informed approach compared to the reference, but not enough to score any h=0 in the hypothesis tests. This suggests that the underlying GANs scheme was particularly bad at reproducing those coefficients. In contrast, the data spread of the out-of-axis coefficients ($c_{1122}^{H}$, $c_{2233}^{H}$, and $c_{1133}^{H}$) in the generated images did not seem to vastly differ from the ground truth, which suggests that physics or thermodynamics central tendency metrics could suffice to improve the model. On the other hand, in the diagonal shear coefficients ($c_{1212}^{H}$, $c_{2323}^{H}$, and $c_{1313}^{H}$), the ground truth plots show a slight positive skewness, which is seemingly not shown by the generated images. This disagreement could be improved by updating the FEM-GANs model with additional metrics using rank sum tests, although there is not much freedom in this regard due to the limited batch sizes.

An empirical cumulative distribution function (CDF) was presented for the nine elasticity coefficients; in this case, the entire database size of 1776 images was used as the ground truth, and 100 image samples were used for the generations (Figure 14). An empirical CDF was used instead of a theoretical CDF because, according to Kolmogorov–Smirnov tests, the data were not normally distributed. The CDFs of the out-of-axis coefficients ($c_{1122}^{H}$, $c_{2233}^{H}$ and $c_{1133}^{H}$) showed a clear mismatch with the ground truth curve, which was in agreement with hypothesis tests. CDF plots allow for visualizing the shape of the distribution tails, as well as presenting the FEM-GANs very good fit curves for the diagonal coefficients. This is otherwise a surprising ability of the model, given the small batch size used during training (eight images).

Both box plots and empirical CDF indicate that the bad performance in the out-of-axis coefficients was a problem inherited from the GANs approach. Possible techniques to mitigate the problem include the following: dynamically adapting the weight of the GANs loss in the informed loss calculation and formulating new equations for the informed *p*-value. The *p*-values and box plots presented in this section can be obtained using the database and code provided at external repository with the DOI: 10.6084/m9.figshare.23280845 (posted on 2 June 2023).

## 5. Conclusions

In this work, the FEM-GANs model has been proposed. The model addresses the existing scarcity of X-ray μCT images for the full-field characterization of materials. This was accomplished by combining a deep learning 3D image generative model with physics supervision using FEM simulations. The main findings are listed below:The loss function computed as the norm of the logarithms was the best performing among the different tested.Results showed that continuous FEM evaluations performed during one additional epoch (with 222 evaluations in total) yielded the best performance.The use of configuration D1 (1 additional epoch, FEM evaluations at every step, 222 FEM evaluations in total, and loss of type norm) is recommended in applications in which only diagonal coefficients are essential.The use of configuration B4 (50 additional epochs, FEM evaluations at every 8 steps, 400 FEM evaluations in total, and loss of type average) is recommended in applications in which out-of-axis coefficients may be important.The model is capable of economically generating endless amounts of images with indistinguishable elastic properties from real images.Generated images have multiple applications, including, but not limited to, data augmentation, calibration and validation of image analysis tools, filling incomplete data in X-ray μCT scans, the generation of variability in statistical methods such as Monte Carlo, and the generation of micro-scales in multi-scale methods.

In future work, the model can be further improved by combining physical information (e.g., the FEM and CFD) with lower levels but computationally cheaper metrics (e.g., fractal, particle count, fabric tensor). The usual purpose of GANs is to obtain a high-quality generator. Nonetheless, the discriminator network can also be used as a classification tool. This use needs to be further explored and validated. The adaptation of the presented approach to state-of-the art GANs can potentially achieve high fidelity images with correct physics, which have applications in a variety of fields.

## Figures and Tables

**Figure 1 materials-16-04740-f001:**
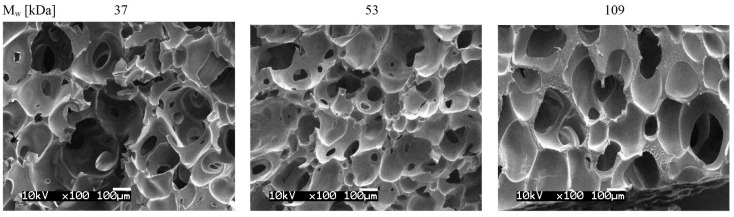
SEM images of scaffolds fabricated by scCO${}_{2}$ foaming: pores and pore windows. Average molecular weights ($M_{w}$) of 37 kDa, 53 kDa, and 109 kDa PLGA (85:15). Adapted with permission from [7].

**Figure 2 materials-16-04740-f002:**
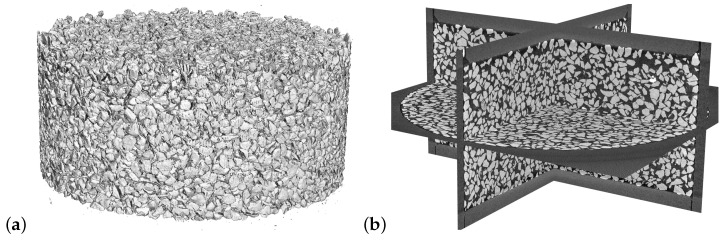
X-ray μCT image of the oedometric test at the loading point with axial stress $\sigma_{z}=500$ kPa and axial strain $\epsilon_{z}=0.019$ [35]. Isosurface with white level threshold 50% (**a**). Slice planes (**b**).

**Figure 3 materials-16-04740-f003:**
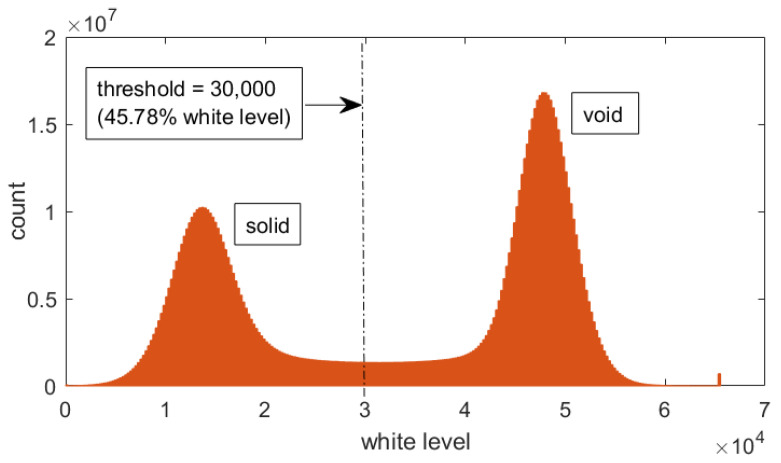
Histogram-based segmentation with sample size 100 and threshold 30,000. White level scale has been reversed to simulate the molded material instead of the sacrifice template. White level range is 1–65,535 (16-bit depth).

**Figure 4 materials-16-04740-f004:**
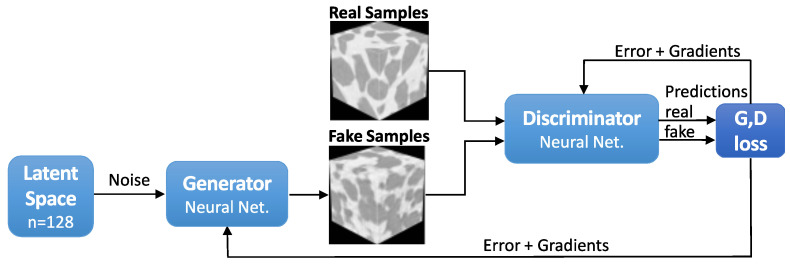
Generative adversarial networks (GANs) schematics. Real samples come from an X-ray μCT image.

**Figure 5 materials-16-04740-f005:**
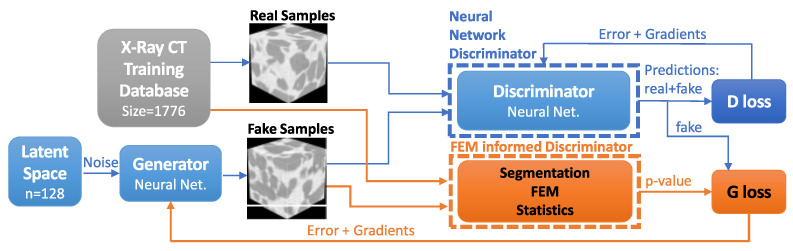
Finite-element-method-informed generative adversarial networks (FEM-GANs) schematics. Blue elements and paths belong to the classic GANs scheme, and orange ones belong to the newly added physics supervision.

**Figure 6 materials-16-04740-f006:**
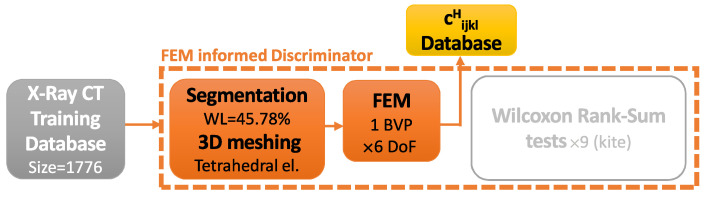
FEM-GANs initial database computation and storage.

**Figure 7 materials-16-04740-f007:**
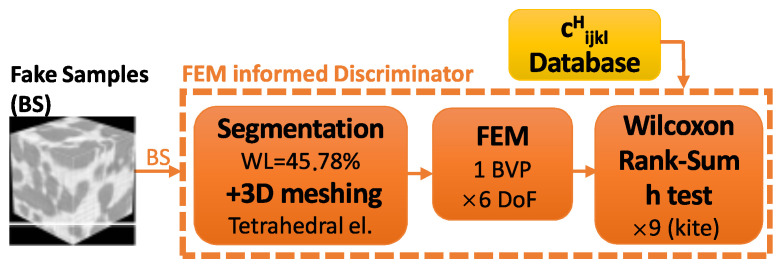
FEM-GANs computed database retrieval during training.

**Figure 8 materials-16-04740-f008:**
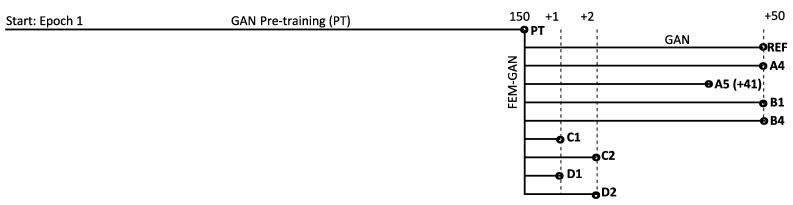
Simulation summary. Pretraining using GANs for 150 epochs. PT serves as a pretrained network for all FEM-GAN simulations and one GANs reference. Epoch lengths not to scale.

**Figure 9 materials-16-04740-f009:**
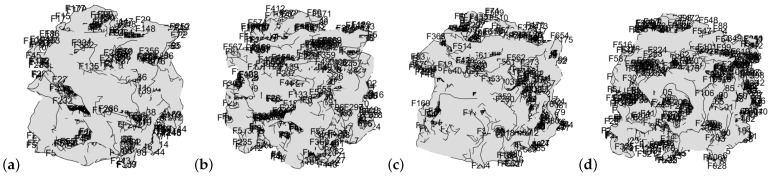
Geometry obtained from the initial meshing using *iso2mesh:* (**a**) Real image. (**b**) Generated image in the pretrained state (epoch 150). (**c**) Generated image in the reference state (epoch 200). (**d**) Image generated by FEM-GANs, simulation D1.

**Figure 10 materials-16-04740-f010:**
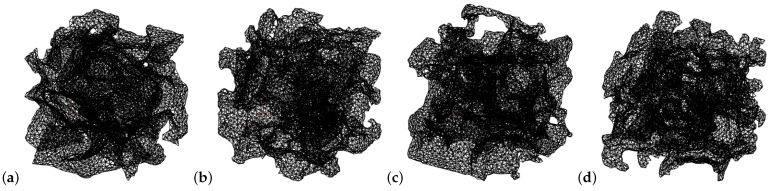
Mesh with tetrahedral elements and quadratic shape functions. (**a**) Real image. (**b**) Generated image in the pretrained state (epoch 150). (**c**) Generated image in the reference state (epoch 200). (**d**) Image generated by FEM-GANs, simulation D1.

**Figure 11 materials-16-04740-f011:**
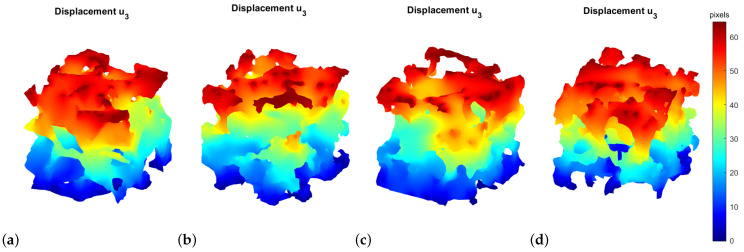
FEM displacement solution for a pure $\epsilon_{33}$ applied boundary strain. (**a**) Real image. (**b**) Generated image in the pre-trained state (epoch 150). (**c**) Generated image in the reference state (epoch 200). (**d**) Generated image by FEM-GANs, simulation D1.

**Figure 12 materials-16-04740-f012:**
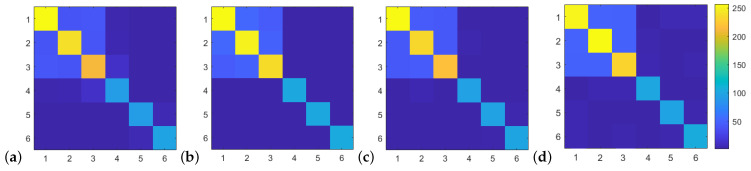
Stiffness tensor coefficient magnitudes exhibiting the typical kite shape generated image. Scale normalized to 8 bits. (**a**) Real image. (**b**) Generated image in the pretrained state (epoch 150). (**c**) Generated image in the reference state (epoch 200). (**d**) Image generated by FEM-GANs, simulation D1.

**Figure 13 materials-16-04740-f013:**
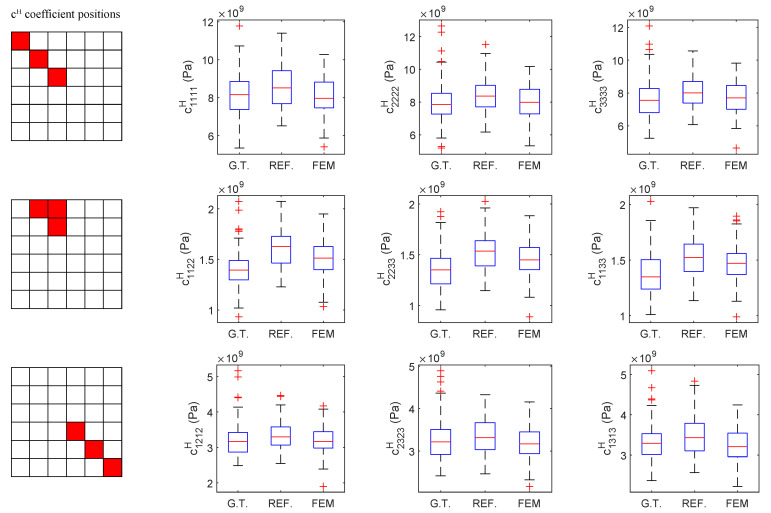
Box plot of the database ground truth (G.T.) images, reference images from GANs (REF.), and images from FEM-informed FEM-GANs model, simulation D1 (FEM). All samples contain 100 images. The horizontal red line indicates the median, blue boxes enclose the interquartile range, black lines are upper and lower adjacents and red crosses outliers.

**Figure 14 materials-16-04740-f014:**
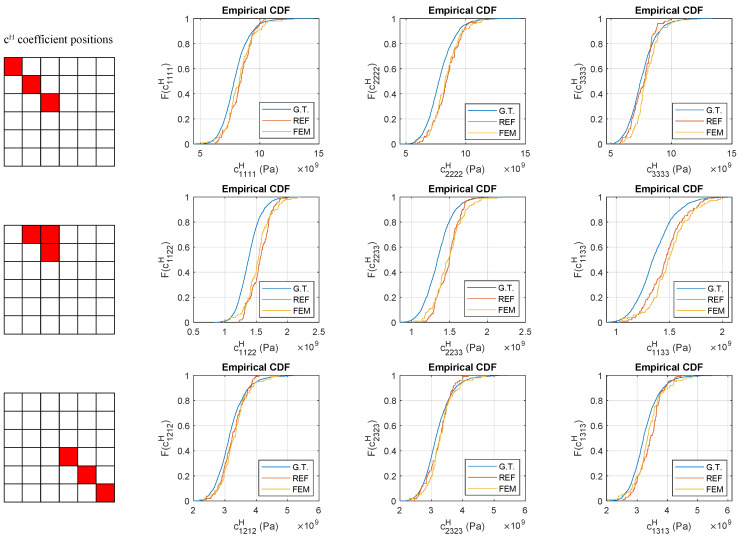
Empirical cumulative distribution function (CDF) plots of ground truth and generated images from reference simulation and FEM-GANs, simulation D1. Ground truth sample size: 1776; generated image sample sizes: 100.

**Table 1 materials-16-04740-t001:** Generator architecture. Output layer with convolution filter kernel size 7×7 and 4th layer with convolution filter kernel size 5×5; all other convolution filters with kernel sizes 3×3.

Layer	Filters	Inference Population	
		Input	Output
1	Fully connected + ReLU + Reshape		128
2	Transposed Convolution + Batch Normalization + ReLU	128	128
3	Transposed Convolution + Batch Normalization + ReLU	128	64
4	Transposed Convolution + Batch Normalization + ReLU	64	32
5	Convolution + Tanh	32	1

**Table 2 materials-16-04740-t002:** Discriminator architecture. Leaky branch slope in Leaky ReLU filters and dropout rate indicated in the parenthesis. Input layer with convolution filter kernel size 7×7 and 2nd layer with convolution filter kernel size 5×5; all other convolution filters with kernel sizes 3×3.

Layer	Filters	Inference Population	
		Input	Output
1	Strided Convolution + Leaky ReLU (leaky slope: 0.2) + Dropout (0.3)	1	16
2	Strided Convolution + Batch Normalization + Leaky ReLU (leaky slope: 0.15) + Dropout (0.3)	16	32
3	Strided Convolution + Batch Normalization + Leaky ReLU (leaky slope: 0.15) + Dropout (0.3)	32	64
4	Strided Convolution + Batch Normalization + Leaky ReLU (leaky slope: 0.15) + Dropout (0.3)	64	128
5	Strided Convolution + Batch Normalization + Leaky ReLU (leaky slope: 0.15) + Dropout (0.3)	128	256
6	Reshape + Fully connected + Sigmoid	256	

**Table 3 materials-16-04740-t003:** FEM-GAN hyperparameter tuning.

Hyperparameter	Value
Latent dimension	128
Batch size	8
Image size	Cubic with 64 voxel side: 0.26 MV ^1^
Discriminator learning rate	$5\times 10^{-5}$
Generator learning rate	$2.5\times 10^{-4}$
Generator decay	0.500
Generator squared decay	0.999
Maximum number of epochs	150 (GANs pre-training) +1/2/50 (FEM-GANs)

^1^ Number of voxels indicated in mega voxels (MV).

**Table 4 materials-16-04740-t004:** Simulation overview. Probability $D_{FEM}\left(G\left(z\right)\right)$ updating strategy is indicated for the simulations not performing FEM evaluations for every iteration.

Parametric Test	Epochs	FEM/Epoch	Iterations with FEM	Loss Function	Probability Updating
pre-training	150	0	0	-	-
REF	50	0	0	-	-
A4	50	8	400	Average Equation (Equation 22)	Persistent
A5	41 ^1^	16	656	Average Equation (Equation 22	Persistent
B1	50	1	50	Average Equation (Equation 22)	Ephemeral
B4	50	8	400	Average Equation (Equation 22)	Ephemeral
C1	1	222 (all)	222	Average Equation (Equation 22)	-
C2	2	222 (all)	444	Average Equation (Equation 22)	-
D1	1	222 (all)	222	Norm Equation (Equation 23)	-
D2	2	222 (all)	444	Norm Equation (Equation 23)	-

^1^ Simulation halted at epoch +41 because of node count error given by the mesher.

**Table 5 materials-16-04740-t005:** Rank sum and *t*-test hypothesis sample sizes.

Images/Sample Size	Parametric	Nonparametric
Real (ground truth)	100	1776 ^1^
Pre-training	100	32
REF	100	32
FEM-GANs	100	32

^1^ Entire database.

**Table 6 materials-16-04740-t006:** Rank sum test *p*-value for each individual elasticity coefficient. 32 image samples. Bold values indicating failure to reject the null hypothesis of equal medians.

Simulation/R-S *p*-Value	$c_{1111}^{H}$	$c_{2222}^{H}$	$c_{3333}^{H}$	$c_{1122}^{H}$	$c_{1133}^{H}$	$c_{2233}^{H}$	$c_{1212}^{H}$	$c_{2323}^{H}$	$c_{1313}^{H}$	h=0 Count
Pre-training	0.0024	0.0007	0.0058	0.0000	0.0000	0.000	0.0339	0.0196	0.0121	0/9
REF	0.0005	0.0052	**0.4563**	0.0000	0.0000	0.0000	**0.1560**	**0.0668**	0.0091	3/9
A4	0.0030	0.0000	**0.0876**	0.0000	0.0000	0.0000	0.0010	0.0542	0.0001	1/9
A5 ^1^	0.0001	0.0000	0.0001	0.0000	0.0000	0.0000	0.0000	0.0001	0.0001	0/9
B1	0.0252	0.0002	0.0035	0.0008	0.0015	0.0172	0.0022	0.0142	0.0062	0/9
B4	**0.2681**	**0.1107**	**0.9742**	0.0001	0.0008	0.0019	**0.7934**	**0.9811**	**0.6058**	6/9
C1	**0.6983**	**0.8898**	0.0378	0.0031	0.0042	0.0001	**0.6265**	**0.3486**	**0.5590**	5/9
C2	0.0105	**0.0921**	**0.5117**	0.0000	0.0018	0.0052	**0.8321**	**0.6850**	0.0327	4/9
D1	**0.6895**	**0.5290**	**0.3090**	0.0002	0.0003	0.0014	**0.7868**	**0.8201**	**0.6598**	6/9
D2	**0.2154**	**0.9285**	**0.1383**	0.0002	0.0070	0.0014	**0.8914**	**0.5450**	**0.9901**	6/9

^1^ Simulation halted at epoch +41 because of node count error given by the mesher.

**Table 7 materials-16-04740-t007:** *t*-test *p*-value for each individual elasticity coefficient. 100 image samples. Bold values indicating failure to reject the null hypothesis of equal means.

Simulation/t-t *p*-Value	$c_{1111}^{H}$	$c_{2222}^{H}$	$c_{3333}^{H}$	$c_{1122}^{H}$	$c_{1133}^{H}$	$c_{2233}^{H}$	$c_{1212}^{H}$	$c_{2323}^{H}$	$c_{1313}^{H}$	h=0 Count
Pre-training	0.0152	0.0064	0.0052	0.0000	0.0000	0.0000	0.0203	**0.0688**	0.0316	1/9
REF	**0.2555**	0.0163	**0.8242**	0.0000	0.0096	0.0000	**0.5815**	**0.7373**	**0.1336**	5/9
A4	**0.0715**	0.0000	0.0425	0.0000	0.0000	0.0000	0.0078	**0.1857**	0.0167	2/9
A5 ^1^	0.0000	0.0000	0.0000	0.0000	0.0000	0.0000	0.0000	0.0000	0.0000	0/9
B1	**0.4248**	0.0287	0.0284	0.0000	0.0001	0.0000	**0.1615**	**0.5320**	**0.4164**	4/9
B4	**0.1491**	**0.5814**	**0.1661**	0.0330	**0.2759**	0.0298	**0.1652**	0.0340	**0.1165**	6/9
C1	**0.2494**	**0.2699**	**0.9172**	0.0073	0.0365	0.0089	**0.1684**	**0.1009**	0.0435	5/9
C2	**0.6958**	**0.3268**	**0.8829**	0.0000	0.0200	0.0029	**0.7372**	**0.3059**	**0.7702**	6/9
D1	**0.5266**	**0.7318**	**0.5983**	0.0000	0.0046	0.0001	**0.9649**	**0.2432**	**0.3394**	6/9
D2	**0.1280**	**0.1649**	0.0339	0.0014	**0.3040**	**0.1478**	0.0150	0.0205	**0.0720**	5/9

^1^ Simulation halted at epoch +41 because of node count error given by the mesher.

## Data Availability

All data available. The FEM-GANs code can be found in the external repository: https://figshare.com/s/12e2a1c7848ddfc0a8d7, DOI: 10.6084/m9.figshare.23280860 (posted on 2 June 2023). Validation can be determined in the external repository: https://figshare.com/s/97a5ddce24c07ec32a67, DOI: 10.6084/m9.figshare.23280845 (posted on 2 June 2023).

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
