# Peer review of "FEM-GAN: A Physics-Supervised Deep Learning Generative Model for Elastic Porous Materials"

_materials, 2023, doi:10.3390/ma16134740_

Round 1

Reviewer 1 Report

I recommend your paper for publication with minimal technical correction.

My remarks relate exclusively to the design of the text. The magazine "Materials" has an extremely simple and convenient template for designing papers https://www.mdpi.com/files/word-templates/applsci-template.dot. Take advantage of it. This will greatly simplify the work of editors and technical proofreaders and decorate your paper.

Notes:

1. In the caption under Figure 1, “mm” must be without a space.

2. A space is required in the numerical values of variables with dimensions, for example, “338 mm” (line 101), “90 GPa” (line 351).

3. In figures 2, 9-12 (a), (b)… “a”, “b”… bold font.

4. Check the correctness of writing all formulas, especially (7), (8).

5. In formulas (5)-(10) the differential is in italics “dA, dS”, and in (11)-(13) the roman type is “dA”. The second option is preferable.

6. Divergence is an operator and it is more logical to write it in plain “div” font. Incidentally, the vector differential operator Ñ (nabla) is related to the divergence div by .

7. All variables in the text of the paper, including figures and tables, for example, in tables 6, 7, should be typed in italics, for example, F, f, N, n, D, d.

8. Carefully look at the rules for the design of bibliography and references to sources. It is desirable to facilitate the process of technical correction of the manuscript. References in the text [1]. MDPI link style:

Chen, J.; Jia, K.; Wang, Z.; Sun, Z. An Intelligent Measurement Method and System for Vehicle Passing Angles. Appl. Sci. 2023, 13, 6677. https://doi.org/10.3390/app13116677

I recommend carefully subtracting the text of the paper in order to eliminate technical errors, maybe I missed something.

Reviewer 2 Report

See my attached referee report.

Reviewer 3 Report

I have thoroughly read the article. My decision is major revision. Kindly see the below comments and revise it:

1. Title of the article needs to be revised…

2. In the abstract of the article, objectives and problem definition written but needs to be revised for the understanding to readers too.

3. Add the more and relevant keywords in the article

4. First paragraph of the introduction section needs to be revised..last two lines meaning is not understanding in first paragraph of introduction section.

5. I found the lumpy references in the introduction section..for a good research article lumpy references must be lower

6. Figures 1 and 2 need to be visible properly.

7. See the table 1 properly

8. How frequently experimentations have been carried out by researchers..

9. See the conclusion of the paper and if possible then add it in bullet form so It will be very helpful for readers.

10. See the future work

11. See all the references properly

Nope

Round 2

Reviewer 3 Report

Article is ready for accepting but please see the below comments:

1. Abstract is still weak...problem definition is not writing properly

2. See the first paragraph, still grammatical errors are there

3. See the conclusion properly

Minor editing
